# Acute Mesenteric Ischemia in COVID-19 Patients

**DOI:** 10.3390/jcm11010200

**Published:** 2021-12-30

**Authors:** Dragos Serban, Laura Carina Tribus, Geta Vancea, Anca Pantea Stoian, Ana Maria Dascalu, Andra Iulia Suceveanu, Ciprian Tanasescu, Andreea Cristina Costea, Mihail Silviu Tudosie, Corneliu Tudor, Gabriel Andrei Gangura, Lucian Duta, Daniel Ovidiu Costea

**Affiliations:** 1Faculty of Medicine, “Carol Davila” University of Medicine and Pharmacy, 020021 Bucharest, Romania; geta.vancea@umfcd.ro (G.V.); ancastoian@yahoo.com (A.P.S.); mihail.tudosie@umfcd.ro (M.S.T.); gabriel.gangura@umfcd.ro (G.A.G.); 2Fourth Surgery Department, Emergency University Hospital Bucharest, 050098 Bucharest, Romania; lulutudor@gmail.com (C.T.); lucian.duta@gmail.com (L.D.); 3Faculty of Dental Medicine, “Carol Davila” University of Medicine and Pharmacy, 020021 Bucharest, Romania; laura.tribus@umfcd.ro; 4Department of Internal Medicine, Ilfov Emergency Clinic Hospital Bucharest, 022104 Bucharest, Romania; 5“Victor Babes” Infectious and Tropical Disease Hospital Bucharest, 030303 Bucharest, Romania; 6Faculty of Medicine, Ovidius University Constanta, 900470 Constanta, Romania; andrasuceveanu@yahoo.com (A.I.S.); Daniel.costea@365.univ-ovidius.ro (D.O.C.); 7Faculty of Medicine, Lucian Blaga University of Sibiu, 550024 Sibiu, Romania; ciprian.tanasescu@ulbsibiu.ro; 8Department of Surgery, Emergency County Hospital Sibiu, 550245 Sibiu, Romania; 9Department of Nephrology, Diaverum Clinic Constanta, 900612 Constanta, Romania; acostea2021@gmail.com; 10Second Surgery Department, Emergency University Hospital Bucharest, 050098 Bucharest, Romania; 11General Surgery Department, Emergency County Hospital Constanta, 900591 Constanta, Romania

**Keywords:** acute mesenteric ischemia, COVID-19, thromboemboembolism, SARS-CoV-2, endothelitis, cytokines, hypercoagulability

## Abstract

Acute mesenteric ischemia is a rare but extremely severe complication of SARS-CoV-2 infection. The present review aims to document the clinical, laboratory, and imaging findings, management, and outcomes of acute intestinal ischemia in COVID-19 patients. A comprehensive search was performed on PubMed and Web of Science with the terms “COVID-19” and “bowel ischemia” OR “intestinal ischemia” OR “mesenteric ischemia” OR “mesenteric thrombosis”. After duplication removal, a total of 36 articles were included, reporting data on a total of 89 patients, 63 being hospitalized at the moment of onset. Elevated D-dimers, leukocytosis, and C reactive protein (CRP) were present in most reported cases, and a contrast-enhanced CT exam confirms the vascular thromboembolism and offers important information about the bowel viability. There are distinct features of bowel ischemia in non-hospitalized vs. hospitalized COVID-19 patients, suggesting different pathological pathways. In ICU patients, the most frequently affected was the large bowel alone (56%) or in association with the small bowel (24%), with microvascular thrombosis. Surgery was necessary in 95.4% of cases. In the non-hospitalized group, the small bowel was involved in 80%, with splanchnic veins or arteries thromboembolism, and a favorable response to conservative anticoagulant therapy was reported in 38.4%. Mortality was 54.4% in the hospitalized group and 21.7% in the non-hospitalized group (*p* < 0.0001). Age over 60 years (*p* = 0.043) and the need for surgery (*p* = 0.019) were associated with the worst outcome. Understanding the mechanisms involved and risk factors may help adjust the thromboprophylaxis and fluid management in COVID-19 patients.

## 1. Introduction

Acute mesenteric ischemia (AMI) is a major abdominal emergency, characterized by a sudden decrease in the blood flow to the small bowel, resulting in ischemic lesions of the intestinal loops, necrosis, and if left untreated, death by peritonitis and septic shock. In non-COVID patients, the etiology may be mesenteric arterial embolism (in 50%), mesenteric arterial thrombosis (15–25%), venous thrombosis (5–15%), or less frequent, from non-occlusive causes associated with low blood flow [1]. Several systemic conditions, such as arterial hypertension, atrial fibrillation, atherosclerosis, heart failure, or valve disease are risk factors for AMI. Portal vein thrombosis and mesenteric vein thrombosis can be seen with celiac disease [2], appendicitis [3], pancreatitis [4], and, in particular, liver cirrhosis and hepatocellular cancer [5].

Acute intestinal ischemia is a rare manifestation during COVID-19 disease, but a correct estimation of its incidence is challenging due to sporadic reports, differences in patients’ selection among previously published studies, and also limitations in diagnosis related to the strict COVID-19 regulations for disease control and difficulties in performing imagistic investigations in the patients in intensive care units. COVID-19 is known to cause significant alteration of coagulation, causing thromboembolic acute events, of which the most documented were pulmonary embolism, acute myocardial infarction, and lower limb ischemia [6].

Gastrointestinal features in COVID-19 disease are relatively frequently reported, varying from less than 10% in early studies from China [7,8] to 30–60%, in other reports [9,10]. In an extensive study on 1992 hospitalized patients for COVID-19 pneumonia from 36 centers, Elmunzer et al. [7] found that the most frequent clinical signs reported were mild and self-limited in up to 74% of cases, consisting of diarrhea (34%), nausea (27%), vomiting (16%), and abdominal pain (11%). However, severe cases were also reported, requiring emergency surgery for acute bowel ischemia or perforation [5,8].

The pathophysiology of the digestive features in COVID-19 patients involves both ischemic and non-ischemic mechanisms. ACE2 receptors are present at the level of the intestinal wall, and enterocytes may be directly infected by SARS-CoV-2. The virus was evidenced in feces and enteral walls in infected subjects [4,11,12,13]. In a study by Xu et al., rectal swabs were positive in 8 of 10 pediatric patients, even after the nasopharyngeal swabs became negative [14]. However, the significance of fecal elimination of viral ARN is still not fully understood in the transmission chain of the SARS-CoV-2 infection. On the other hand, disturbance of lung-gut axis, prolonged hospitalization in ICU, and the pro coagulation state induced by SARS-CoV-2 endothelial damage was incriminated for bowel ischemia, resulting in intestinal necrosis and perforation [8,9,15]. Early recognition and treatment of gastrointestinal ischemia are extremely important, but it is often challenging in hospitalized COVID-19 patients with severe illness.

The present review aims to document the risk factors, clinical, imagistic, and laboratory findings, management, and outcomes of acute intestinal ischemic complications in COVID-19 patients.

## 2. Materials and Methods

A comprehensive search was performed on PubMed and Web of Science with the terms “COVID-19” AND (“bowel ischemia” OR “intestinal ischemia” OR “mesenteric ischemia” OR “mesenteric thrombosis”). All original papers and case reports, in the English language, for which full text could be obtained, published until November 2021, were included in the review. Meeting abstracts, commentaries, and book chapters were excluded. A hand search was performed in the references of the relevant reviews on the topic.

### 2.1. Data Extraction and Analysis

The review is not registered in PROSPERO. A PRISMA flowchart was employed to screen papers for eligibility (Figure 1) and a PRISMA checklist is presented as a Appendix A. A data extraction sheet was independently completed by two researchers, with strict adherence to PRISMA guidelines.

The relevant data abstracted from these studies are presented in Table 1, Table 2 and Table 3. COVID-19 diagnosis was made by PCR assay in all cases. All patients reported with COVID-19 disease and mesenteric ischemia were documented in terms of age, sex, comorbidities, time from SARS-CoV-2 infection diagnosis, presentation, investigations, treatment, and outcome. A statistical analysis of the differences between acute intestinal ischemia in previously non-hospitalized vs. previously hospitalized patients was performed. The potential risk factors for an adverse vital prognosis were analyzed using SciStat^®^ software (www.scistat.com (accessed on 25 November 2021)).

Papers that did not provide sufficient data regarding evaluation at admission, documentation of SARS-CoV-2 infection, or treatment were excluded. Patients suffering from other conditions that could potentially complicate intestinal ischemia, such as liver cirrhosis, hepatocellular carcinoma, intraabdominal infection (appendicitis, diverticulitis), pancreatitis, and celiac disease were excluded. Any disagreement was solved by discussion.

### 2.2. Risk of Bias

The studies analyzed in the present review were comparable in terms of patient selection, methodology, therapeutic approach, and the report of final outcome. However, there were differences in the reported clinical and laboratory data. The sample size was small, most of them being case reports or case series, which may be a significant source of bias. Therefore, studies were compared only qualitatively.

## 3. Results

After duplication removal, a total of 36 articles were included in the review, reporting data on a total of 89 patients. Among these, we identified 6 retrospective studies [16,17,18,19,20,21], documenting intestinal ischemia in 55 patients admitted to intensive care units (ICU) with COVID-19 pneumonia for whom surgical consult was necessary (Table 1).

We also identified 30 case reports or case series [22,23,24,25,26,27,28,29,30,31,32,33,34,35,36,37,38,39,40,41,42,43,44,45,46,47,48,49,50,51] presenting 34 cases of acute bowel ischemia in patients positive for SARS-CoV-2 infection in different clinical settings. 8 cases were previously hospitalized for COVID-19 pneumonia and under anticoagulant medication (Table 2). In 26 cases, the acute ischemic event appeared as the first symptom of COVID-19 disease, or in mild forms treated at home, or after discharge for COVID -19 pneumonia and cessation of the anticoagulant medication (Table 3).

### 3.1. Risk Factors of Intestinal Ischemia in COVID-19 Patients

Out of a total of 89 patients included in the review, 63 (70.7%) were hospitalized for severe forms of COVID-19 pneumonia at the moment of onset. These patients were receiving anticoagulant medication when reported, consisting of low molecular weight heparin (LMWH) at prophylactic doses. The incidence of acute intestinal ischemia in ICU patients with COVID-19 varied widely between 0.22–10.5% (Table 1). In a study by O’Shea et al. [20], 26% of hospitalized patients for COVID-19 pneumonia who underwent imagistic examination, presented results positive for coagulopathy, and in 22% of these cases, the thromboembolic events were with multiple locations.

The mean age was 56.9 years. We observed a significantly lower age in non-hospitalized COVID-19 patients presenting with acute intestinal ischemia when compared to the previously hospitalized group (*p* < 0.0001).

There is a slight male to female predominance (M:F = 1:68). Obesity might be considered a possible risk factor, with a reported mean BMI of 31.2–32.5 in hospitalized patients [16,18,19]. However, this association should be regarded with caution, since obesity is also a risk factor for severe forms of COVID-19. Prolonged stay in intensive care, intubation, and the need for vasopressor medication was associated with increased risk of acute bowel ischemia [8,18,19].

Diabetes mellitus and hypertension were the most frequent comorbidities encountered in case reports (8 in 34 patients, 23%), and 7 out of 8 patients presented both (Table 4). There was no information regarding the comorbidities in the retrospective studies included in the review.

### 3.2. Clinical Features in COVID-19 Patients with Acute Mesenteric Ischemia

Abdominal pain, out of proportion to physical findings, is a hallmark of porto-mesenteric thrombosis, typically associated with fever and leukocytosis [4]. Abdominal pain was encountered in all cases, either generalized from the beginning, of high intensity, or firstly localized in the epigastrium or the mezogastric area. In cases of portal vein thrombosis, the initial location may be in the right hypochondrium, mimicking biliary colic [26,34].

Fever is less useful in COVID-19 infected patients, taking into consideration that fever is a general sign of infection, and on the other hand, these patients might be already under antipyretic medication.

Other clinical signs reported were nausea, anorexia, vomiting, and food intolerance [23,31,38,45]. However, these gastrointestinal signs are encountered in 30–40% of patients with SARS-CoV-2 infection. In a study by Kaafarani et al., up to half of the patients with gastrointestinal features presented some degrees of intestinal hypomotility, possibly due to direct viral invasion of the enterocytes and neuro-enteral disturbances [16].

Physical exam evidenced abdominal distension, reduced bowel sounds, and tenderness at palpation. Guarding may be evocative for peritonitis due to compromised vascularization of bowel loops and bacterial translocation or franc perforation [35,39].

A challenging case was presented by Goodfellow et al. [25] in a patient with a recent history of bariatric surgery with Roux en Y gastric bypass, presenting with acute abdominal pain which imposed the differential diagnosis with an internal hernia.

Upcinar et al. [24] reported a case of an 82-years female that also associated atrial fibrillation. The patient was anticoagulated with enoxaparin 0.4 cc twice daily before admission and continued the anticoagulant therapy during hospitalization for COVID-19 pneumonia. Bedside echocardiography was performed to exclude atrial thrombus. Although SMA was reported related to COVID-19 pneumonia, atrial fibrillation is a strong risk factor for SMA of non-COVID-19 etiology.

In ICU patients, acute bowel ischemia should be suspected in cases that present acute onset of digestive intolerance and stasis, abdominal distension, and require an increase of vasopressor medication [19].

### 3.3. Imagistic and Lab Test Findings

D-dimer is a highly sensitive investigation for the prothrombotic state caused by COVID-19 [45] and, when reported, was found to be above the normal values. Leukocytosis and acute phase biomarkers, such as fibrinogen and CRP were elevated, mirroring the intensity of inflammation and sepsis caused by the ischemic bowel. However, there was no significant statistical correlation between either the leukocyte count (*p* = 0.803) or D-dimers (*p* = 0.08) and the outcome. Leucocyte count may be within normal values in case of early presentation [34]. Thrombocytosis and thrombocytopenia have been reported in published cases with mesenteric ischemia [30,35,42,46,50].

Lactate levels were reported in 9 cases, with values higher than 2 mmol/L in 5 cases (55%). LDH was determined in 6 cases, and it was found to be elevated in all cases, with a mean value of 594+/−305 U/L.

Ferritin is another biomarker of potential value in mesenteric ischemia, that increases due to ischemia-reperfusion cellular damage. In the reviewed studies, serum ferritin was raised in 7 out of 9 reported cases, with values ranging from 456 to 1570 ng/mL. However, ferritin levels were found to be correlated also with the severity of pulmonary lesions in COVID-19 patients [52]. Due to the low number of cases in which lactate, LDH, and ferritin were reported, no statistical association could be performed with the severity of lesions or with adverse outcomes.

The location and extent of venous or arterial thrombosis were determined by contrast-enhanced abdominal CT, which also provided important information on the viability of the intestinal segment whose vascularity was affected.

Radiological findings in the early stages included dilated intestinal loops, thickening of the intestinal wall, mesenteric fat edema, and air-fluid levels. Once the viability of the affected intestinal segment is compromised, a CT exam may evidence pneumatosis as a sign of bacterial proliferation and translocation in the intestinal wall, pneumoperitoneum due to perforation, and free fluid in the abdominal cavity. In cases with an unconfirmed diagnosis of COVID-19, examination of the pulmonary basis during abdominal CT exam can add consistent findings to establish the diagnosis.

Venous thrombosis affecting the superior mesenteric vein and or portal vein was encountered in 40.9% of reported cases of non-hospitalized COVID-19 patients, and in only one case in the hospitalized group (Table 5). One explanation may be the beneficial role of thrombotic prophylaxis in preventing venous thrombosis in COVID-19 patients, which is routinely administrated in hospitalized cases, but not reported in cases treated at home with COVID-19 pneumonia.

In ICU patients, CT exam showed in most cases permeable mesenteric vessels and diffuse intestinal ischemia affecting the large bowel alone (56%) or in association with the small bowel (24%), suggesting pathogenic mechanisms, direct viral infection, small vessel thrombosis, or “nonocclusive mesenteric ischemia” [16].

### 3.4. Management and Outcomes

The management of mesenteric ischemia includes gastrointestinal decompression, fluid resuscitation, hemodynamic support, anticoagulation, and broad antibiotics.

Once the thromboembolic event was diagnosed, heparin, 5000IU iv, or enoxaparin or LMWH in therapeutic doses was initiated, followed by long-term oral anticoagulation and/or anti-aggregating therapy. Favorable results were obtained in 7 out of 9 cases (77%) of splanchnic veins thrombosis and in 2 of 7 cases (28.5%) with superior mesenteric artery thrombosis. At discharge, anticoagulation therapy was continued either with LMWH, for a period up to 3 months [33,36,41], either, long term warfarin, with INR control [32,34,41] or apixaban 5 mg/day, up to 6 months [26,47]. No readmissions were reported.

Antibiotic classes should cover anaerobes including *F. necrophorum* and include a combination of beta-lactam and beta-lactamase inhibitor (e.g., piperacillin-tazobactam), metronidazole, ceftriaxone, clindamycin, and carbapenems [4].

In early diagnosis, during the first 12 h from the onset, vascular surgery may be tempted, avoiding the enteral resection [25,53]. Endovascular management is a minimally invasive approach, allowing quick restoration of blood flow in affected vessels using techniques such as aspiration, thrombectomy, thrombolysis, and angioplasty with or without stenting [40].

Laparotomy with resection of the necrotic bowel should be performed as quickly as possible to avoid perforation and septic shock. In cases in which intestinal viability cannot be established with certainty, a second look laparotomy was performed after 24–48 h [43] or the abdominal cavity was left open, using negative pressure systems such as ABTHERA [51], and successive segmentary enterectomy was performed.

Several authors described in acute bowel ischemia encountered in ICU patients with COVID-19, a distinct yellowish color, rather than the typical purple or black color of ischemic bowel, predominantly located at the antimesenteric side or circumferentially with affected areas well delineated from the adjacent healthy areas [18,19]. In these cases, patency of large mesenteric vessels was confirmed, and the histopathological reports showed endothelitis, inflammation, and microvascular thrombosis in the submucosa or transmural. Despite early surgery, the outcome is severe in these cases, with an overall mortality of 45–50% in reported studies and up to 100% in patients over 65 years of age according to Hwabejira et al. [19].

In COVID-19 patients non hospitalized at the onset of an acute ischemic event, with mild and moderate forms of the disease, the outcome was less severe, with recovery in 77% of cases.

We found that age over 60 years and the necessity of surgical treatment are statistically correlated with a poor outcome in the reviewed studies (Table 6). According to the type of mesenteric ischemia, the venous thrombosis was more likely to have a favorable outcome (recovery in 80% of cases), while vascular micro thombosis lead to death in 66% of cases.

## 4. Discussions

Classically, acute mesenteric ischemia is a rare surgical emergency encountered in the elderly with cardiovascular or portal-associated pathology, such as arterial hypertension, atrial fibrillation, atherosclerosis, heart failure, valve disease, and portal hypertension. However, in the current context of the COVID-19 pandemic, mesenteric ischemia should be suspected in any patient presenting in an emergency with acute abdominal pain, regardless of age and associated diseases.

Several biomarkers were investigated for the potential diagnostic and prognostic value in acute mesenteric ischemia. Serum lactate is a non-specific biomarker of tissue hypoperfusion and undergoes significant elevation only after advanced mesenteric damage. Several clinical trials found a value higher than 2 mmol/L was significantly associated with increased mortality in non-COVID-patients. However, its diagnostic value is still a subject of debate. There are two detectable isomers, L-lactate, which is a nonspecific biomarker of anaerobic metabolism, and hypoxia and D-lactate, which is produced by the activity of intestinal bacteria. Higher D-lactate levels could be more specific for mesenteric ischemia due to increased bacterial proliferation at the level of the ischemic bowel, but the results obtained in different studies are mostly inconsistent [53,54].

Several clinical studies found that LDH is a useful biomarker for acute mesenteric ischemia, [55,56]. However, interpretation of the results may be difficult in COVID-19 patients, as both lactate and LDH were also found to be independent risk factors of severe forms of COVID-19 [57,58].

The diagnosis of an ischemic bowel should be one of the top differentials in critically ill patients with acute onset of abdominal pain and distension [50,59]. If diagnosed early, the intestinal ischemia is potentially reversible and can be treated conservatively. Heparin has an anticoagulant, anti-inflammatory, endothelial protective role in COVID-19, which can improve microcirculation and decrease possible ischemic events [25]. The appropriate dose, however, is still a subject of debate with some authors recommending the prophylactic, others the intermediate or therapeutic daily amount [25,60].

We found that surgery is associated with a severe outcome in the reviewed studies. Mucosal ischemia may induce massive viremia from bowel epithelium causing vasoplegic shock after surgery [25]. Moreover, many studies reported poor outcomes in COVID-19 patients that underwent abdominal surgery [61,62].

### 4.1. Pathogenic Pathways of Mesenteric Ischemia in COVID-19 Patients

The intestinal manifestations encountered in SARS-CoV-2 infection are represented by inflammatory changes (gastroenteritis, colitis), occlusions, ileus, invaginations, and ischemic manifestations. Severe inflammation in the intestine can cause damage to the submucosal vessels, resulting in hypercoagulability in the intestine. Cases of acute cholecystitis, splenic infarction, or acute pancreatitis have also been reported in patients infected with SARS-CoV-2, with microvascular lesions as a pathophysiological mechanism [63].

In the study of O’Shea et al., on 146 COVID-19 hospitalized patients that underwent CT-scan, vascular thrombosis was identified in 26% of cases, the most frequent location being in lungs [20]. Gastrointestinal ischemic lesions were identified in 4 cases, in multiple locations (pulmonary, hepatic, cerebellar parenchymal infarction) in 3 patients. The authors raised awareness about the possibility of underestimation of the incidence of thrombotic events in COVID-19 patients [20].

Several pathophysiological mechanisms have been considered, and they can be grouped into occlusive and non-occlusive causes [64]. The site of the ischemic process, embolism or thrombosis, may be in the micro vascularization, veins, or mesenteric arteries.

Acute arterial obstruction of the small intestinal vessels and mesenteric ischemia may appear due to hypercoagulability associated with SARS-CoV-2 infection, mucosal ischemia, viral dissemination, and endothelial cell invasion vis ACE-2 receptors [65,66]. Viral binding to ACE2Receptors leads to significant changes in fluid-coagulation balance: reduction in Ang 2 degradation leads to increased Il6 levels, and the onset of storm cytokines, such as IL-2, IL-7, IL-10, granulocyte colony-stimulating factor, IgG -induced protein 10, monocyte chemoattractant protein-1, macrophage inflammatory protein 1-alpha, and tumor necrosis factor α [67], but also in the expression of the tissue inhibitor of plasminogen -1, and a tissue factor, and subsequently triggering the coagulation system through binding to the clotting factor VIIa [68]. Acute embolism in small vessels may be caused by the direct viral invasion, via ACE-2 Receptors, resulting in endothelitis and inflammation, recruiting immune cells, and expressing high levels of pro-inflammatory cytokines, such as Il-6 and TNF-alfa, with consequently apoptosis of the endothelial cells [69].

Capillary viscometry showed hyperviscosity in critically ill COVID-19 patients [70,71]. Platelet activation, platelet–monocyte aggregation formation, and Neutrophil external traps (NETs) released from activated neutrophils, constitute a mixture of nucleic DNA, histones, and nucleosomes [59,72] were documented in severe COVID-19 patients by several studies [70,71,73].

Plotz et al. found a thrombotic vasculopathy with histological evidence for lectin pathway complement activation mirroring viral protein deposition in a patient with COVID-19 and SLE, suggesting this might be a potential mechanism in SARS-CoV-2 associated thrombotic disorders [47].

Numerous alterations in fluid-coagulation balance have been reported in patients hospitalized for COVID-19 pneumonia. Increases in fibrinogen, D-dimers, but also coagulation factors V and VIII. The mechanisms of coagulation disorders in COVID-19 are not yet fully elucidated. In a clinical study by Stefely et al. [68] in a group of 102 patients with severe disease, an increase in factor V > 200 IU was identified in 48% of cases, the levels determined being statistically significantly higher than in non-COVID mechanically ventilated or unventilated patients hospitalized in intensive care. This showed that the increased activity of Factor V cannot be attributed to disease severity or mechanical ventilation. Additionally, an increase in factor X activity was shown, but not correlated with an increase in factor V activity, but with an increase in acute phase reactants, suggesting distinct pathophysiological mechanisms [74].

Giuffre et al. suggest that fecal calcoprotein (FC) may be a biomarker for the severity of gastrointestinal complications, by both ischemic and inflammatory mechanisms [75]. They found particularly elevated levels of FC to be well correlated with D-dimers levels in patients with bowel perforations, and hypothesized that the mechanism may be related to a thrombosis localized to the gut and that FC increase is related to virus-related inflammation and thrombosis-induced ischemia, as shown by gross pathology [76].

Non-occlusive mesenteric ischemia in patients hospitalized in intensive care units for SARS-CoV-2 pneumonia requiring vasopressor medication may be caused vasospastic constriction [19,64,65]. Thrombosis of the mesenteric vessels could be favored by hypercoagulability, relative dehydration, and side effects of corticosteroids.

### 4.2. Question Still to Be Answered

Current recommendations for in-hospital patients with COVID-19 requiring anticoagulation suggest LMWH as first-line treatment has advantages, with higher stability compared to heparin during cytokine storms, and a reduced risk of interaction with antiviral therapy compared to oral anticoagulant medication [77]. Choosing the adequate doses of LMWH in specific cases—prophylactic, intermediate, or therapeutic—is still in debate. Thromboprophylaxis is highly recommended in the absence of contraindications, due to the increased risk of venous thrombosis and arterial thromboembolism associated with SARS-CoV-2 infection, with dose adjustment based on weight and associated risk factors. Besides the anticoagulant role, some authors also reported an anti-inflammatory role of heparin in severe COVID-19 infection [66,78,79]. Heparin is known to decrease inflammation by inhibiting neutrophil activity, expression of inflammatory mediators, and the proliferation of vascular smooth muscle cells [78]. Thromboprophylaxis with enoxaparin could be also recommended to ambulatory patients with mild to moderate forms of COVID-19 if the results of prospective studies show statistically relevant benefits [80].

In addition to anticoagulants, other therapies, such as anti-complement and interleukin (IL)-1 receptor antagonists, need to be explored, and other new agents should be discovered as they emerge from our better understanding of the pathogenetic mechanisms [81]. Several studies showed the important role of Il-1 in endothelial dysfunction, inflammation, and thrombi formation in COVID-19 patients by stimulating the production of Thromboxane A2 (TxA2) and thromboxane B2 (TxB2). These findings may justify the recommendation for an IL-1 receptor antagonist (IL-1Ra) which can prevent hemodynamic changes, septic shock, organ inflammation, and vascular thrombosis in severe forms of COVID-19 patients [80,81,82].

## 5. Conclusions

Understanding the pathological pathways and risk factors could help adjust the thromboprophylaxis and fluid management in COVID-19 patients. The superior mesenteric vein thrombosis is the most frequent cause of acute intestinal ischemia in COVID-19 non-hospitalized patients that are not under anticoagulant medication, while non-occlusive mesenteric ischemia and microvascular thrombosis are most frequent in severe cases, hospitalized in intensive care units.

COVID-19 patients should be carefully monitored for acute onset of abdominal symptoms. High-intensity pain and abdominal distension, associated with leukocytosis, raised inflammatory biomarkers and elevated D-dimers and are highly suggestive for mesenteric ischemia. The contrast-enhanced CT exam, repeated, if necessary, offers valuable information regarding the location and extent of the acute ischemic event. Early diagnosis and treatment are essential for survival.

## Figures and Tables

**Figure 1 jcm-11-00200-f001:**
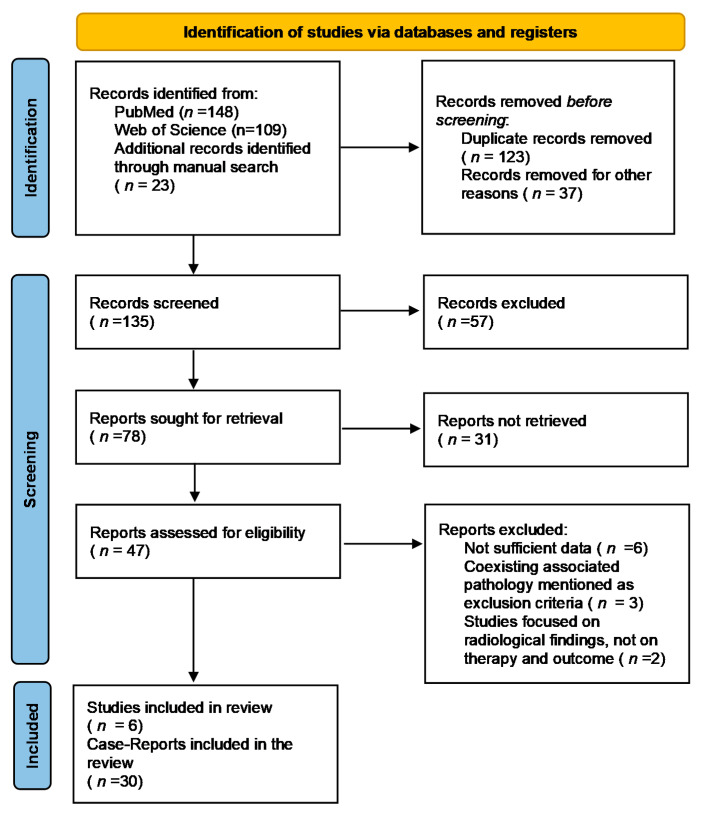
PRISMA 2020 flowchart for the studies included in the review.

**Table 1 jcm-11-00200-t001:** Patients with intestinal ischemia in retrospective studies on hospitalized COVID-19 patients.

Study	No of Patients with Gastrointestinal Ischemia (Total No of COVID-19 Patients in ICU)	Sex (M; F)	Age (Mean)	BMI	Time from Admission to Onset (Days)	Abdominal CT Signs	Intraoperative/Endoscopic Findings	Treatment	Outcomes
Kaafarani HMA [16]	5 (141); 3.8%	1;3	62.5	32.1	51.5 (18–104) days	NA	Cecum-1—patchy necrosis Cecum_ileon-1 Small bowel-3; yellow discoloration on the antimesenteric side of the small bowel; 1 case + liver necrosis	Surgical resection	NA
Kraft M [17]	4 (190); 2.1%	NA	NA	NA	NA	NA	Bowel ischemia + perforation (2) Bowel ischemia + perforation (1) MAT+massive bowel ischemia (1)	Right hemicolectomy (2) Transverse colectomy (1) Conservative, not fit for surgery	Recovery (3) Death (1)
Yang C [18]	20 (190 in ICU; 582 in total); 10.5%	15:5	69	31.2	26.5 (17–42)	Distension Wall thickness Pneumatosis intestinalis Perforation SMA or celiac thrombosis	no info	Right hemicolectomy 7(35%) Sub/total colectomy12 (60%) Ileocecal resection 1(5%)	Recovery (11) Death (9)
Hwabejire J [19]	20	13:7	58.7	32.5	13 (1–31)	Pneumatosis intestinalis 42% Portal venous gas (33%) Mesenteric vessel patency 92%	large bowel ischemia (8) small bowel ischemia (4) both (8) yellow discoloration of the ischemic bowel	resection of the ischemic segment abdomen left open + second look (14)	Recovery (10) Death (10)
O’Shea A [20]	4 (142); 2.8%	NA	NA	NA	NA	bowel ischemia, portal vein gas, colic pneumatosis	NA	NA	NA
Qayed E [21]	2 (878); 0.22%	NA	NA	NA	NA	NA	diffuse colonic ischemia (1) Small + large bowel ischemia and pneumatosis (1)	Total colectomy (1) Extensive resection (1)	Recovery (1) Death (1)

NA: not acknowledged; MAT: mesenteric artery thrombosis; SMA: superior mesenteric artery.

**Table 2 jcm-11-00200-t002:** Case reports and case series presenting gastrointestinal ischemia in hospitalized COVID-19 patients under anticoagulant medication.

Article	Sex	Age	Comorbidities	Time from COVID-19 Diagnosis; Time from Admission (Days)	ICU; Type of Ventilation	Clinical Signs at Presentation	Leukocytes (/mm^3^)	CRP (mg/L)	Lactat mmol/L	Ferritin (ng/mL)	LDH (U/L)	Thrombocytes (/mm3)	D-Dimers (ng/mL)	Abdominal CT Signs	Treatment	Outcome
Azouz E [22]	M	56	none	1; 2 (hospitalized for acute ischemic stroke)	No info	abdominal pain and vomiting	No info	-	-	-	-	-	-	Multiple arterial thromboembolic complications: AMS, right middle cerebral artery, a free-floating clot in the aortic arch	Anticoagulation (no details), endovascular thrombectomy Laparotomy + resection of necrotic small bowel loops	No info
Al Mahruqi G [23]	M	51	none	26; 24	yes, intubated	Fever, metabolic acidosis, required inotropes	30,000	-	7	687	-	-	2.5	Non-occlusive AMI Hypoperfused small bowel, permeable aorta, SMA, IMA + deep lower limb thrombosis	enoxaparin 40 mg/day from admission; surgery refused by family	death
Ucpinar BA [24]	F	82	Atrial fibrillation, hypertension, chronic kidney disease	3; 3	no	-	14,800	196	5.1	-	-	-	1600	SMA thrombosis; distended small bowel, with diffuse submucosal pneumatosis portomesenteric gas	fluid resuscitation; continued ceftriaxone, enoxaparin 0.4cc twice daily; not operable due to fulminant evolution	Death
Karna ST [25]	F	61	DM, hypertension	4; 4	Yes, HFNO	diffuse abdominal pain with distention	21,400	421.6	1.4	-	-	464,000	No	thrombosis of the distal SMA with dilated jejunoileal loops and normal enhancing bowel wall.	Iv heparin 5000 ui, followed by 1000 ui, Ecospin and clopidogrel Laparotomy after 10 days with segmental enterectomy of the necrotic bowel	Death by septic shock and acute renal failure
Singh B [26]	F	82	Hypertension, T2DM	32; 18	Yes, Ventilator support	severe diffuse abdominal distension and tenderness	22,800	308	2.5	136	333	146,000	1.3	SMA—colic arteries thrombosis pneumatosis intestinalis affecting the ascending colon and cecum	laparotomy, ischemic colon resection, ileostomy; heparin in therapeutic doses pre- and post-surgery	slow recovery
Nakatsutmi K [27]	F	67	DM, diabetic nephropathy requiring dialysis, angina, post-resection gastric cancer	16; 12	ICU, intubation	hemodynamic deterioration, abdominal distension	15,100	32.14	-	-	-	-	26.51	edematous transverse colon; abdominal vessels with sclerotic changes	laparotomy, which revealed vascular micro thrombosis of transverse colon—right segment resection of the ischemic colonic segment, ABTHERA management, second look, and closure of the abdomen after 24 h	death
Dinoto E [28]	F	84	DM, hypertension, renal failure	2; 2	no	Acute abdominal pain and distension;	18,000	32.47	-	-	431	-	6937	SMA origin stenosis and occlusion at 2 cm from the origin, absence of bowel enhancement	Endovascular thrombectomy of SMA; surgical transfemoral thrombectomy and distal superficial femoral artery stenting	Death due to respiratory failure
Kiwango F [29]	F	60	DM, hypertension	12; 3	no	Sudden onset abdominal pain	7700	-	-	-	-	-	23.8	Not performed	Not performed due to rapid oxygen desaturation Massive bowel acute ischemia	death

**Table 3 jcm-11-00200-t003:** Case reports and case series presenting gastrointestinal ischemia in non-hospitalized COVID-19 patients.

Article	Sex	Age	Comorbidities	Time from COVID-19 Diagnosis (Days)	Clinical Signs at Presentation	Leukocyte Count (/mm^3^)	CRP (mg/L)	Lactate mmol/L	Ferritin (ng/mL)	LDH (U/L)	Thrombocytes (/mm^3^)	D-Dimers (ng/mL)	Abdominal CT Signs	Treatment	Outcome
Sevella, P [30]	M	44	none	10	Acute abdominal pain constipation, vomiting	23,400	-	-	-	1097	360,000	1590	Viable jejunum, ischemic bowel, peritoneal thickening with fat stranding; free fluid in the peritoneal cavity	LMWH 60 mg daily Piperacillin 4g/day Tazobactam 500 mg/day Extensive small bowel + right colon resection	death
Nasseh S [31]	M	68	no info	First diagnosis	epigastric pain and diarrhea for 4 days	17,660	125	-	-	-	-	6876	terminal segment of the ileocolic artery thrombosis; thickening of the right colon wall and the last 30 cm of the small bowl	unfractionated heparin laparoscopy -no bowel resection needed	recovery
Aleman W [32]	M	44	none	20	severe abdominopelvic pain	36,870	-	-	456.23	-	574,000	263.87	absence of flow at SMV, splenic, portal vein; Small bowel loop dilatation and mesenteric fat edema	enoxaparin and pain control medication 6 days, then switched to warfarin 6 months	recovery
Jeilani M [33]	M	68	Alzheimer disease, COPD	9	Sharp abdominal pain +distension	12,440	307	-	-	-	318,000	897	a central venous filling defect within the portal vein extending to SMV; no bowel wall changes	LMWH, 3 months	recovery
Randhawa J [34]	F	62	none	First diagnosis	right upper quadrant pain and loss of appetite for 14 days	Normal limits	-	-	-	346	-	-	large thrombus involving the SMV, the main portal vein with extension into its branches	Fondaparinux 2.5. mg 5 days, then warfarin 4 mg (adjusted by INR), 6 months	recovery
Cheung S [35]	M	55	none	12 (discharged for 7 days)	Nausea, vomiting and worsening generalized abdominal pain with guarding	12,446	-	0.68	-	-	-	-	low-density clot, 1.6 cm in length, causing high-grade narrowing of the proximal SMA	continuous heparin infusion continued 8 h postoperative, Laparotomy with SMA thromboembolectomy and enterectomy (small bowel)	recovery
Beccara L [36]	M	52	none	22 (5 days after discharge and cessation prophylactic LWMH)	vomiting and abdominal pain, tenderness in epigastrium and mesogastrium	30,000	222	-	-	-	-	-	arterial thrombosis of vessels efferent of the SMA with bowel distension	Enterectomy (small bowel) LMWH plus aspirin 100 mg/day at discharge	recovery
Vulliamy P [37]	M	75	none	14	abdominal pain and vomiting for 2 days	18,100	3.2	-	-	-	497,000	320	intraluminal thrombus was present in the descending thoracic aorta with embolic occlusion of SMA	Catheter-directed thrombolysis, enterectomy (small bowel)	recovery
De Barry O [38]	F	79	none	First diagnosis	Epigastric pain, diarrhea, fever for 8 days, acute dyspnea	12600	125	5.36	-	-	-	-	SMV, portal vein, SMA, and jejunal artery thrombosis Distended loops, free fluid	anticoagulation Resection of affected colon+ ileum, SMA thrombolysis, thrombectomy	death
Romero MCV [39]	M	73	smoker, DM, hypertension	14	severe abdominal pain, nausea. fecal emesis, peritoneal irritation	18,000	-	-	-	-	120,000	>5000	RX: distention of intestinal loops, inter-loop edema, intestinal pneumatosis	enoxaparin (60 mg/0.6 mL), antibiotics (no info) enterectomy, anastomotic fistula, reintervention	death
Posada Arango [40]	M F F	62 22 65	None Appendectomy 7 days before left nephrectomy,	5 3 15	colicative abdominal pain at food intake; unsystematized gastrointestinal symptoms; abdominal pain in the upper hemiabdomen	20,100 - -	- - -	- - -	1536 - -	534 - -	- - -	- - -	Case 1: thrombus in distal SMA and its branches, intestinal loops dilatation, hydroaerical levels, free fluid thrombosis of SMV Case 2: SMV thrombosis and adiacent fat edema Case 3: thrombi in the left jejunal artery branch with infarction of the corresponding jejunal loops	Case 1: Laparotomy: extensive jejunum + ileum ischemia; surgery could not be performed Case 2: Anticoagulation analgesic and antibiotics Case 3: segmental enterectomy	Case 1: death Case 2: recovery Case 3: recovery
Pang JHQ [41]	M	30	none	First diagnosis	colicky abdominal pain, vomiting	-	-	-	-	-	-	20	SMV thrombosis with diffuse mural thickening and fat stranding of multiple jejunal loops	conservative, anticoagulation with LMWH 1mg/kc, twice daily, 3 months; readmitted and operated for congenital adherence causing small bowel obstruction	recovery
Lari E [42]	M	38	none	First diagnosis	abdominal pain, nausea, intractable vomiting, and shortness of breath	Mild leukocytosis	-	2.2	-	-	-	2100	extensive thrombosis of the portal, splenic, superior, and inferior mesenteric veins + mild bowel ischemia	Anticoagulation, resection of the affected bowel loop	No info
Carmo Filho A [43]	M	33	Obesity (BMI: 33), other not reported	7	severe low back pain radiating to the hypogastric region	-	58.2	-	1570	-	-	879	enlarged inferior mesenteric vein not filled by contrast associated with infiltration of the adjacent adipose planes	enoxaparin 5 days, followed by long term oral warfarin	recovery
Hanif M [44]	F	20	none	8	abdominal pain and abdominal distension	15,900	62	-	1435.3	825	633,000	2340	not performed	evidence of SMA thrombosis; enterectomy with exteriorization of both ends	recovery
Amaravathi U [45]	M	45	none	5	Acute epigastric and periumbilical pain	-	Normal value	1.3	324.3	-	-	5.3	SMA and SMV thrombus	i.v. heparin; Laparotomy with SMA thrombectomy; 48 h Second look: resection of the gangrenous bowel segment	No info
Al Mahruqi G [23]	M	51	none	4	generalized abdominal pain, nausea, vomiting	16,000	-	-	619	-	-	10	SMA thrombosis and non-enhancing proximal ileal loops consistent with small bowel ischemia	unfractionated heparin, thrombectomy + repeated resections of the ischemic bowel at relook (jejunum+ileon+cecum)	Case 2: recovery
Goodfellow M [46]	F	36	RYGB, depression, asthma	6	epigastric pain, irradiating back, nausea	9650	1.2	0.7	-	-	-	-	abrupt cut-off of the SMV in the proximal portion; diffuse infiltration of the mesentery, wall thickening of small bowel	IV heparin infusion, followed by 18,000 UI delteparin after 72 h	recovery
Abeysekera KW [26]	M	42	Hepatitis B	14	right hypochondrial pain, progressively increasing for 9 days	-	-	-	-	-	-	-	enhancement of the entire length of the portal vein and a smaller thrombus in the mid-superior mesenteric vein, mural edema of the distal duodenum, distal small bowel, and descending colon	factor Xa inhibitor apixaban 5 mg ×2/day, 6 months	-recovery
Rodriguez-Nakamura RM [27]	M F	45 42	-vitiligo -obesity	14	severe mesogastric pain, nausea, diaphoresis	16,400 18,800	367 239	- -	970 -	- -	685,000 -	1450 14,407	Case 1: SMI of thrombotic etiology with partial rechanneling through the middle colic artery, and hypoxic-ischemic changes in the distal ileum and the cecum Case 2: thrombosis of the portal and mesenteric veins and an abdominopelvic collection in the mesentery with gas	Case 1: resection with entero-enteral anastomosis; rivaroxaban 10 mg/day, 6 months Case 2: Loop resection, entero-enteral manual anastomosis, partial omentectomy, and cavity wash (fecal peritonitis)	Case 1: Recovery Case 2: death
Plotz B [47]	F	27	SLE with ITP	First diagnosis	acute onset nausea, vomiting, and non-bloody diarrhea	-	-	-	-	-	-	5446	diffuse small bowel edema	enoxaparin, long term apixaban at discharge	recovery
Chiu CY [48]	F	49	Hypertension, DM, chronic kidney disease	28	diffuse abdominal pain melena and hematemesis	-	-	-	-	-	-	12,444	distended proximal jejunum with mural thickening	laparotomy, proximal jejunum resection	no info
Farina D [49]	M	70	no info	3	abdominal pain, nausea	15,300	149	-	-	-	-	-	acute small bowel hypoperfusion, SMA thromboembolism	not operable due to general condition	Death

SMA: superior mesenteric artery; SMV: superior mesenteric vein; DM: diabetes mellitus; T2DM: type 2 diabetes mellitus; AMI: acute mesenteric ischemia; IMV: inferior mesenteric vein; RYGB: Roux-en-Y gastric bypass (bariatric surgery).

**Table 4 jcm-11-00200-t004:** Demographic data of the patients included in the review.

Nr. of Patients	89
M	48 (61.5% *)
F	30 (38.5% *)
NA	11
The first sign of COVID-19	6 (6.7%)
Home treated	17 (19.1%)
Hospitalized ICU	63 (70.7%) 58 (92% of hospitalized patients)
Discharged	3 (3.3%)
Time from diagnosis of COVID-19 infection Non-Hospitalized Hospitalized (*when mentioned)	8.7 ± 7.4 (1–28 days) 9.6 ± 8.3 (1–26 days)
Time from admission in hospitalized patients	1–104 days
Age (mean) Hospitalized Non-hospitalized	59.3 ± 12.7 years 62 ± 9.6 years. (*p* < 0.0001) 52.8 ± 16.4 years.
BMI	31.2–32.5
Comorbidities Hypertension DM smokers Atrial fibrillation COPD Cirrhosis RYGB Vitiligo Recent appendicitis Operated gastric cancer Alzheimer disease SLE	8 7 2 1 2 1 1 1 1 1 1 1

*: percentage calculated in known information group; BMI: body mass index; COPD: chronic obstructive pulmonary disease; SLE: systemic lupus erythematosus.

**Table 5 jcm-11-00200-t005:** Comparative features in acute intestinal ischemia encountered in previously hospitalized and previously non-hospitalized COVID-19 patients.

Parameter	Hospitalized (63)	Non-Hospitalized (26)	*p* * Value
Type of mesenteric ischemia: Arterial Venous Mixt (A + V) Diffuse microthrombosis Multiple thromboembolic locations NA	5 (14.7% *) 1 (2.9%) 0 30 (88.2%) 2 (5.8%) 29	10 (38.4%) 11 (42.3%) 2 (7.6%) 3 (11.5%) 1 (3.8%) 0	*p* < 0.0001
Management: Anticoagulation therapy only Endovascular thrombectomy Laparotomy with ischemic bowel resection None (fulminant evolution)	0 2 (1 + surgery) (3%) 60 (95.4%) 2 (3%)	10 (38.4%) 2 (+surgery) 15 (57.6%) 1 (3.8%)	*p* < 0.0001
Location of the resected segment: Colon Small bowel Colon+small bowel NA	35 (56%) 10 (16%) 15 (24%) 6	0 12 (80%) 3 (20%) 0	*p* < 0.0001
Outcomes: Recovery Death NA	26 (46.4%) 30 (54.4%) 7	17 (79.3%) 5 (21.7%) 3	*p* = 0.013

* calculated for Chi-squared test.

**Table 6 jcm-11-00200-t006:** Risk factors for severe outcome.

Parameters	Outcome: Death	*p*-Value
Age Age < 60 Age > 60	27.2% 60%	0.0384 * 0.043 **
Surgery No surgery surgery	0% 60%	0.019 **
Type of mesenteric ischemia Arterial Venous Micro thrombosis	47% 20% 66%	0.23 **
D dimers	Wide variation	0.085 * 0.394 **
Leucocytes	Wide variation (9650–37,000/mmc)	0.803 0.385 **

* One-way ANOVA test; ** Chi-squared test (SciStat^®^ software, www.scistat.com (accessed on 25 November 2021)).

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
