# Peer review of "Acute Mesenteric Ischemia in COVID-19 Patients"

_jcm, 2021, doi:10.3390/jcm11010200_

Round 1

Reviewer 1 Report

Topic treated is interesting and strongly current. However, there are too many tables and a diagram is missing that illustrates the review criteria followed by the authors (eg PRISMA statement).

Author Response

Dear reviewer,

Thank you very much for your time and effort in reviewing our manuscript and for your valuable comments.

Response: We have restructured the manuscript as a systematic review, following the PRISMA guidelines. We detailed the search strategy and the inclusion and exclusion criteria in the Materials and Methods section. We added a PRISMA flowchart in the Material and methods section.

We revised the tables, by condensing the information and adding the data asked by the other reviewer regarding LDH, ferritin, lactate levels and thrombocytes. To be easier to follow one patient’s evolution from admission to final outcome, we condensed the information in only one table for the case reports admitted for mesenteric ischemia (table 3; these patients were not taking anticoagulant medication at admission) and one table (table 2) for the patients admitted for covid-19 pneumonia, who developed mesenteric ischemia during hospitalization, despite anticoagulant prophylaxis.

We could move tables 2 and 3 in the supplementary materials if you consider them appropriate. However, please consider that in the JCM format, the tables are presented as icons, that may be opened separately, so they do not impact the fluidity of the main text.

We statistically analyzed the correlation between different factors and the final outcome, based on the available data in the case reports (table 6), as recommended by the other reviewer.

We do hope that in this revised version you will find it suitable to be published

Kind regards,

Dr. Ana Dascalu

Reviewer 2 Report

I have read with interest and very carefully this review article about mesenteric thrombosis in patients diagnosed with COVID 19. There is a lot to like about this paper, and if accepted it will contribute to existing literature on this emerging topic. The idea is novel and there are no many publications about COVID 19 and thrombosis of mesenteric and portal vein as its related to COVID 19. This being said, I do find several major points, that in my opinion need to be completely addressed before the paper can be re-considered.

Abstract:

  1. You cannot determine the incidence of the disease or complication by this type of systematic reviews. This is not a prospective study where such indicator can be measured. Sence, the incidence needs to be removed throughout the paper.
  2. Paraclinical features? What are those? To my knowledge this terminology in English doesn’t exist or at least is not common in the literature- I would suggest changing to laboratory and imaging findings
  3. Abstract needs to be re-written and include the findings of the study. Instead of “constant findings”- specific percentage should be mentioned. Additionally, “leucocyte” are constant finding doesn’t make sense- authors likely wanted to say leucocytosis

Introduction:

  1. The lines 45-52 are common knowledge, almost every paper starts with this. It is boring. Particularly useless are the numbers of infected and deaths as by the time paper is published this will already change. I would suggest deleting this part and replace it with mesenteric ischemia and etiology of it other than COVID. For example, we know that portal vein thrombosis and mesenteric vein thrombosis can be seen with celiac disease ( https://pubmed.ncbi.nlm.nih.gov/29379656/), also with appendicitis ( https://pubmed.ncbi.nlm.nih.gov/26793462/) , pancreatitis ( https://pubmed.ncbi.nlm.nih.gov/25513905/), and in particular with liver cirrhosis and hepatocellular cancer ( https://bmcgastroenterol.biomedcentral.com/articles/10.1186/1471-230X-7-34). This would be much more interesting and pertinent for the study you conducted, yet will be very educational for the readers

Methodology:

  1. This type of reviews- systematic reviews need to be followed by PRISMA guidelines. Narrative reviews are invited from the experts in the field and, as such they do not need to follow these guidelines. In this review, it is mandatory that authors follow up PRISMA and report findings, including PRISMA diagram accordingly
  2. LDH, lactate levels, and ferritin would be of interest. Lactate particularly since it is a very useful marker for bowel ischemia. If able, authors should report these.
  3. Clinical manifestation- abdominal pain is hallmark of porto-mesenteric thrombosis =. As one review of pyelophlebitis found- all patients had fever, abdominal pain and leucocytosis ( https://pubmed.ncbi.nlm.nih.gov/31993242/). Authors reported about leucocytosis, fever will not be very useful as patients had a COVID and fever is a common sign of infection. However, abdominal pain % should be reported and more discussion should be centered about this.
  4. It would be important to know if the thrombosis occurred on admission or later in the course of the disease.
  5. Authors should also specifically report if other conditions ( from introduction ) such as liver cirrhosis, HCC, intraabdominal infection( appendicitis, diverticulitis), pancreatitis, celiac disease where excluded. This is particularly important as some of the patients had SLE , gastric cancer, ruptured appendicitis, RA, cirrhosis etc
  6. Table where you report co-morbidities there are two rows with SLE each with one patient- consolidate to SLE-2

Results:

  1. In one table ( 4) you report 2 deaths and in other (Table 1) only 2. Please explain/clarify.
  2. It would be important to evaluate if any factors such as age>65, D dimer above certain limit, CRP, etc is associated with mortality as it seem that almost 25% of patients died. I understand this might not be possible if you don’t have details from the retrospective studies you included. However, you might need to run analysis only on patients from case reports and case studies
  3. One patient in you study had atrial fibrillation. As you are aware embolic complications are common with A fib. Please comment on this- was the patient anticoagulated- how was systemic embolization ruled out?

Treatment

  1. Duration of anticoagulation should be reported. Duration and choice ( if available ) of antibiotics.

Other comments:

  1. Line 184- symptoms suggestive of? Please be more specific
  2. Line 186- broad antibiotherapy should be changed to broad antibiotics, or antibiotic therapy or antimicrobial therapy
  3. Line 295- IL-1 or IL-6?

Author Response

Dear reviewer,

Thank you very much for your time and effort in reviewing our manuscript and for your valuable comments. Here we present in detail how we responded to every issue:

Abstract:

  1. You cannot determine the incidence of the disease or complication by this type of systematic review. This is not a prospective study where such indicator can be measured. Sence, the incidence needs to be removed throughout the paper.

Response: We have removed it.

  1. Paraclinical features? What are those? To my knowledge this terminology in English doesn’t exist or at least is not common in the literature- I would suggest changing to laboratory and imaging findings

Response: we have rephrased  as recommended

  1. Abstract needs to be re-written and include the findings of the study. Instead of “constant findings”- specific percentage should be mentioned. Additionally, “leucocyte” are constant finding doesn’t make sense- authors likely wanted to say leucocytosis

Response: We have made the required changes. 

Introduction:

  1. The lines 45-52 are common knowledge, almost every paper starts with this. It is boring. Particularly useless are the numbers of infected and deaths as by the time paper is published this will already change. I would suggest deleting this part and replace it with mesenteric ischemia and etiology of it other than COVID. For example, we know that portal vein thrombosis and mesenteric vein thrombosis can be seen with celiac disease ( https://pubmed.ncbi.nlm.nih.gov/29379656/), also with appendicitis ( https://pubmed.ncbi.nlm.nih.gov/26793462/) , pancreatitis ( https://pubmed.ncbi.nlm.nih.gov/25513905/), and in particular with liver cirrhosis and hepatocellular cancer ( https://bmcgastroenterol.biomedcentral.com/articles/10.1186/1471-230X-7-34). This would be much more interesting and pertinent for the study you conducted, yet will be very educational for the readers

Response: We have modified the introduction, based on your suggestions and references.

Methodology:

  1. This type of reviews- systematic reviews need to be followed by PRISMA guidelines. Narrative reviews are invited from the experts in the field and, as such they do not need to follow these guidelines. In this review, it is mandatory that authors follow up PRISMA and report findings, including PRISMA diagram accordingly.

Response: We have restructured the manuscript as a systematic review, following the PRISMA guidelines. We detailed the search strategy and the inclusion and exclusion criteria in the Materials and Methods section. We added a PRISMA flowchart in the Material and methods section.

2. LDH, lactate levels, and ferritin would be of interest. Lactate particularly since it is a very useful marker for bowel ischemia. If able, authors should report these.

Response: we added in the tables all available data regarding LDH, ferritin, lactat and thrombocytes.

To be easier to follow the case reports from admission to final outcome, we condensed the information in only one table for the case reports admitted for mesenteric ischemia (table 3; these patients were not taking anticoagulant medication at admission) and one table (table 2) for the patients admitted for covid-19 pneumonia, who developed mesenteric ischemia during hospitalization, despite anticoagulant prophylaxis.

3. Clinical manifestation- abdominal pain is hallmark of porto-mesenteric thrombosis =. As one review of pyelophlebitis found- all patients had fever, abdominal pain and leucocytosis ( https://pubmed.ncbi.nlm.nih.gov/31993242/). Authors reported about leucocytosis, fever will not be very useful as patients had a COVID and fever is a common sign of infection. However, abdominal pain % should be reported and more discussion should be centered about this.

Response: we have detailed the section regarding the clinical findings in acute mesenteric ischemia, as recommended

4. It would be important to know if the thrombosis occurred on admission or later in the course of the disease.

Response: For the clinical studies, the mesenteric ischemia was reported during hospitalization (table 1, column 6). To be clearer in presentation, we separate the case reports by this criterion: in table 2 we present the cases that were hospitalized for covid -19 pneumonia and mesenteric ischemia appeared during hospitalization, while these patients were already on anticoagulant medication (LMWH – prophylactic doses). In table 2, column 5, we added information regarding the time lapsed from the moment of Covid-19 diagnosis/the time from hospital admission.

In table 3, we present cases that presented for mesenteric ischemia at the emergency department. In some cases, the Sars-Cov-2 infection was ignored by the patients and found at admission; in other cases, there were mild forms of disease treated at home, until the acute event. None of these patients were on anticoagulant medication at admission.

5. Authors should also specifically report if other conditions ( from introduction ) such as liver cirrhosis, HCC, intraabdominal infection( appendicitis, diverticulitis), pancreatitis, celiac disease where excluded. This is particularly important as some of the patients had SLE , gastric cancer, ruptured appendicitis, RA, cirrhosis etc

Response: We added a paragraph in the material and methods, describing the exclusion criteria. All active conditions that could have as complication intestinal ischemia were excluded from the review

6. Table where you report co-morbidities there are two rows with SLE each with one patient- consolidate to SLE-2

Response: only one reported patient had SLE, we have corrected the information in the table.

Results:

  1. In one table ( 4) you report 2 deaths and in other (Table 1) only 2. Please explain/clarify.

Response: We have revised and corrected this error.

2. It would be important to evaluate if any factors such as age>65, D dimer above certain limit, CRP, etc is associated with mortality as it seem that almost 25% of patients died. I understand this might not be possible if you don’t have details from the retrospective studies you included. However, you might need to run analysis only on patients from case reports and case studies

Thank you for the interesting suggestion. We add a statistical analysis of the risk factors associated with mortality based on the available data in Table 6.

3. One patient in your study had atrial fibrillation. As you are aware embolic complications are common with A fib. Please comment on this- was the patient anticoagulated- how was systemic embolization ruled out?

Response: The authors of the cited report (Upcinar et al) mentioned that: “bedside echocardiography was performed to exclude atrial thrombus; and nothing remarkable was observed except for mild cardiomegaly”. The patient was anticoagulated with enoxaparin 0.4 cc twice daily before admission and continued the anticoagulant therapy during hospitalization for Covid-19 pneumonia. However, although SMA was reported related to Covid-19 pneumonia, atrial fibrillation is a strong risk factor for SMA of non-Covid-19 etiology

Treatment

  1. Duration of anticoagulation should be reported. Duration and choice ( if available ) of antibiotics.

Response: We added all the available info regarding the anticoagulation and antibiotic therapy in the table 2 and 3, “Treatment” section. We also added a paragraph in the Treatment and oucome section, regarding the length and options of anticoagulant therapy at discharge.

Other comments:

  1. Line 184- symptoms suggestive of? Please be more specific

Response: we have rephrased

2. Line 186- broad antibiotherapy should be changed to broad antibiotics, or antibiotic therapy or antimicrobial therapy

Response: we have replaced with broad antibiotics

3. Line 295- IL-1 or IL-6?

Response: Il-1. We added a short explanatory paragraph with the deleterious effects of Il-1 that may result in thrombi formation

We do hope that in this revised version you will find it suitable to be published. 

Kind regards,

Dr. Ana Dascalu

Round 2

Reviewer 2 Report

I would like to thank the authors for a very detailed revision of the paper. All of my concerns have been appropriately addressed. The paper has been significantly improved, and I have no further question. In my opinion it should be accepted and published as such.